# An OCARI-Based Wireless Sensor Network for Heat Measurements during Outdoor Fire Experiments

**DOI:** 10.3390/s19010158

**Published:** 2019-01-04

**Authors:** Thierry Carlotti, Xavier Silvani, Eric Innocenti, Frédéric Morandini, Nicolas Bulté, Tuan Dang

**Affiliations:** 1Laboratoire Sciences Pour l’Environnement, CNRS UMR 6134-University of Corsica, Bat.CRIT 3ieme etage, Campus Grimaldi, 20 250 CORTE Northern Corsica, France; carlotti@univ-corse.fr (T.C.); ino@univ-corse.fr (E.I.); morandin@univ-corse.fr (F.M.); 2Société Insulaire Pétrolière Lieu-dit Baléone, 20167 Sarolla-Carcopino Southern Corsica, France; Nicolas.bulte@entreprise-sip.fr; 3Electricite de France-R&D, Department STEP, Control Systems and Information Technologies Group, 78401 Chatou CEDEX, France; tuan.dang@edf.fr

**Keywords:** WSN experiment, outdoor fire of natural fuel, heat measurement

## Abstract

This study presents a new wireless measurement system for outdoor fire experiments based on an OCARI wireless sensor network (WSN). The open-source radio communication OCARI stack (*‘Open Communication Protocol for Ad Hoc Reliable Industrial Instrumentation’*) allows to overcome the limitations of the previous wireless solutions for fire measurements, especially those based on proprietary systems, such as Zigbee, wherein users cannot control energy consumption and timestamping. This paper presents the design of an Atmel/ARM-based platform compatible with the wireless OCARI stack, adapted to record and communicate data from heat transducers—namely, a K-type thermocouple and heat fluxmeter. A 5-node WSN using heat sensors faces a 10 m${}^{2}$ natural fire of excelsior (pine wood) in outdoor conditions. Measurements from the OCARI-WSN were compared with the same measurements simultaneously recorded on a datalogger, a standard wired solution in fire science. Thus, six fire experiments were performed for different fuel loads, and the incident heat radiation varies with the fire size under variable wind conditions. The technological breakdown of timestamping identified in previous low-energy consumption and low-cost wireless systems is overcome by the present solution.

## 1. Introduction

Large-scale measurements are indispensable prerequisites aimed at improving and validating physics-based models of natural gas confined fires. Typical measurement devices needed for instrumenting fire phenomena must be adapted to outdoor conditions [1], especially when transducers are very close to or embedded in a fire [2]. The main objectives of these metrological developments involve capturing both long temporal trends of large-scale phenomena and rapid fluctuations of physical quantities as heat or chemical quantities [3,4]. Furthermore, it is difficult to set and display the fire metrology on a field scale, owing to the financial costs for installing devices over large areas and to maintain the metrological integrity facing large-scale fires. In this framework, measurement systems including transducers, datalogger, and their wires for large-scale fire experiments may benefit from the use of wireless sensor network systems (WSNs) [5]. Specifically, the coverage of wide areas initially requires a significant quantity of wiring to connect transducers to the datalogger, which has a significant financial impact on the overall solution costs. However, wires are particularly vulnerable when facing a fire, while maintaining prolonged contact with it, or when they are exposed to long-range heat fluxes. These wires must therefore be protected with heat shields or/and should be buried. These requirements increase the overall cost of experiments. In industrial applications where fire detection or measurement is required, wires must also ensure safe operating conditions, which contributes to making this solution very expensive. Therefore, in fire science and fire safety, wire costs are estimated to range from 300 euros/m in natural conditions to 6000 euros/m in explosive industrial environments [6]. We discuss these points in the following section; however, the technical and functional specifications of future radio communication protocols require the communication system to be autoconfigurable and resilient (able to repair itself) and exhibit low energy consumption working without human intervention. This expected autonomy also improves cost effectiveness in comparison to a wired solution.

WSN are wireless solutions for networks which have to present two strong properties. Contrary to Wi-Fi, which requires an infrastructure to work, i.e., one or several routers and a network architecture to plan before, WSN are ad hoc: Radio communications can be set up at each node of network, which behaves as a send–receive router. The route of the digital information, i.e., the way it flows from a point to another is recomputed periodically and dynamically. That ensures that the WSN network is autoconfigurable whatever the circumstances of its display. A second skill proceeds from this ad hoc feature of the network: It is resilient, i.e., it is able to autorepair. Indeed, if a node in the routing map breaks down, its neighbours will recompute a new route for transmitting the data at the next cycle. Only a few examples of a solution based on a WSN architecture in experiment detection and monitoring systems for fire are available in literature [7,8,9,10,11]. However, up to now, in the limit of our knowledge of the scientific literature about WSN and fire, none of these WSN solutions has already been used in order to perform continuous measurements over the area to monitor. These are merely designed for fire detection.

A key question is, therefore, whether replacing the standard set—i.e., a network of analogue transducers connected by wire to a datalogger—with a WSN-based solution leads to reliable measurements of heat quantities for fire monitoring and mitigation. This must correspond to a sufficient quality comparable to standard wired solutions. A discussion by the authors of [6] compares WSN-based and wired heat measurements during fire experiments. In this previous study, a wireless system that uses a Zigbee radio communication protocol and implements heat transducers was compared to a wired sensor system during fire experiments performed under several indoor and outdoor conditions. This is the first experiment involving a WSN solution for performing heat measurement during a fire scenario, i.e., more than sole fire detection. Despite this ambition, this study mainly illustrated the limitations of a commercial Zigbee generic protocol, which does not allow for convenient timestamping of data on each wireless node, thus introducing an important time delay between the wireless system and its wired reference. Further, this study also revealed a loss of radio messages when the measurement point was embedded in the flame front, probably due to the interaction between radio electromagnetic waves and the fire environment [12]. The resolution of these limitations suffers from the fact that WSNs mainly correspond to proprietary systems, even if they are based on the IEEE 802.15.4 protocols (Zigbee, Wireless Hart, ISA100, etc.) for MC-based architectures. In these proprietary systems, a specific design of the hardware or software part of the solution is not always possible such that all their parameters (at the application and network layers) vary to optimally match user needs. These close-to-real time requirements and efficient data timestamping in measurement systems form challenging key points in every computer network dedicated to operating in time-critical scenarios [13], such as floods, landslip wasting, exposure to toxic gases, and evidently every possible fire scenario in an industrial explosive environment or in the open [14].

Designed at the end of the 2010s, *‘Open Communication Protocol for Ad Hoc Reliable Industrial Instrumentation’*) (OCARI) appears to be a modern alternative to the usual limitations of wireless commercial protocols based on the IEEE 802.15.4, for MC-based sensing solutions. The OCARI stack is open-source, and each user can develop a few specific applications based on an industrial-quality and long-term maintained source.

In the study, we propose a WSN-based measurement solution to monitor outdoor fires based on an OCARI protocol [15]. The aim of the solution involves overcoming technological breakdowns identified with the Zigbee-based WSN solution for fire measurements provided in Reference [6]. Specifically, in the study, we evaluate the ability of an OCARI-based heat measurement system to solve technological limitations of incorrect timestamping and intermessage delay observed in previous cases with the Zigbee fire solution.

The remainder of this paper is organised as follows: In the second section, we briefly present the OCARI protocol for industrial wireless sensor networks and explain how its features may solve the limitations observed with the Zigbee one in fire experiments (Section 2); Section 3 presents measurements devices, the wired datalogger, and the experimental setup of the heat series. Section 4 deals with the results and the discussion before the conclusion (Section 5).

## 2. Wireless Sensors Networks (WSN) in the IEEE 802.15.4: Zigbee and OCARI Protocols

IEEE 802.15.4 establishes most of the requirements allowing the development of efficient and powerful WSN devices [16]. It specifies three networking topologies, namely star, tree, and meshed, as shown in Figure 1. These skills are important to identify if the network solution is adapted to the area subject to fire hazard. This section describes the principles of IEEE 802.15.4 protocols in terms that are adapted to individuals who are not familiar with a radio communication network, and it presents new advantages of OCARI when compared to the Zigbee ones.

### 2.1. General Information

OCARI and ZigBee Technologies are designed for use in a short-range wireless personal area network (WPAN) context. These technologies are typically used to deploy dedicated wireless networks for data transmission between home devices or in industrial environments. They have shared a growing interest since the 2000s, with the home automation market, which has evolved significantly since 2003, leading to the Internet of Things (IoT) market, which is still moving today. These kind of networks, particularly WSNs, are of great interest to the scientific community [17]. Specifically, WSN devices are based on WPAN technologies in which a few standards are commonly used in daily life (for example, Bluetooth). A few others (HomeRF, ZigBee, LiFi, etc.) are increasingly used for the collection of environmental data, especially given their many advantages in terms of costs, performance, and usability. The performance of these devices is closely related to protocols in charge of physical and MAC layers of the wireless network and the deepest layers of the hardware–software architecture of the network elements.

#### 2.1.1. Related Works in Fire Studies

Before introducing each stack in order to exhibit the advantages of OCARI vs. Zigbee, let us first observe that there is no previous attempt to perform continuous measurements through a WSN strategy in the frame of fire studies. Indeed, the present study aims to break the technological limitation of the incorrect timestamping of data transmitted for measurements versus a real scale fire using a WSN. Even if it is non-exhaustive, the list [7,8,9,10,11] illustrates that the literature only reports on wireless sensors systems for fire detection in the open or in harsh industrial environments. The continuous measurements of physical quantities (heat, air content of moisture) is rarely addressed with a wireless sensor network. The reasons must be explained in this paragraph. The IEEE 802.15.4 radio communication protocol for microcontroller-based networks can work according two modes: As presented in Figure 1, the nodes can either be set to work as a full function device (FFD), i.e., each node is able to route the digital information towards accessible neighbours, or they work as end-devices (reduced function device in Figure 1), just sending a message to a local FFD. The FFD mode is energy-consuming because routes are frequently recomputed by the stack for each network. A measurement protocol necessarily requires this mode because the digital information must continuously flow through the network despite a constantly possible breakdown of a node in the routing protocol. However, in an FFD configuration, this wireless measurement solution is energy-consuming, as more of the radio environment is polluted by external perturbations than in a fire. This is why in practical cases, and except our tests in Reference [6] with a Zigbee-based solution, there is no paper illustrating the use of IEEE 802.15.4 for measurements during fire experiments. In Reference [6], the energy consumption by node approaches 1 Ah/day, in a star topology with a Zigbee-based heat measurement system. That illustrates the IEEE 802.15.4 can only be set up as an FFD for short-time measurement campaigns (about 1 day) and that the long-term autonomy of the Zigbee network imposes an RFD parametrisation of the network. The list [7,8,9,10,11] reports on fire monitoring with a WSN solution where the protocol only sends radio packets when a fire causes the measurements to change over certain values [7] or for fire detection only [8,9,10,11]: Both concern the RFD configuration of the node network contrary to ours in Reference [6].

### 2.2. OCARI Protocol Stack

In this part, we describe the main OCARI principles that retain our attention for this work. The stack denotes the vertical structure of software that includes several functions at different levels of abstraction. These levels are called layers. A network stack is, therefore, the overall programmed instructions from the hardware layer up to the software one, containing instructions for network communications. More complete OCARI descriptions can be found in References [18,19].

#### 2.2.1. General Information

The OCARI was initially a French research project that was funded by the French National Research Agency (ANR) [20]. This project started in December 2006, and it ended in November 2009 [21]. The project gathered industrial partners (DCNS, EDF, and Telit) and academic partners (INRIA, LATTIS, LIMOS, and LRI). The main objective of the project was to develop WSN technologies that can operate in strongly constrained environments and specifically in industrial environments. Specifically, OCARI is currently implemented on a platform that is based on the Atmel IEEE 802.15.4 transceiver and the ARM Cortex M3 microcontroller. This type of a platform is typically commercialised by Dresden Electronik GmbH, Adwave, and HiKoB. Figure 2 shows where OCARI technologies are situated when compared to ZigBee technologies, from which they were largely inspired. When Zigbee is an upper layer of the IEEE 802.15.4 layer, OCARI technologies optimally stand out from the ZigBee ones: In particular, they introduce a new medium access control layer protocol called MaCARI, which is in charge of enhancing medium use, and a specific scheduling algorithm, which allows to improve network activity, called OSERENA, which is used in its updated version. These are succinctly described in Appendix A for the sake of readability.

The OCARI protocol stack thus competes with the IEEE 802.15.4 technologies dedicated to WSNs, and in particular with ZigBee technologies. The OCARI protocol stack offers an opportunity to easily deploy meshed networks at a large scale in an industrial environment or at a field scale for experimental measurements. OCARI offers significantly greater performances in energy consumption than Zigbee. The theoretical limit of the operable modules is approximately 400 nodes. It is necessary for an OCARI network to deploy two types of nodes, namely a network controller (also termed as CPAN) and PAN modules (which are fully functional devices as stated in IEEE 802.15.4 terminology) that form the network nodes equipped with sensors or actuators. The working principle of an OCARI network is illustrated in Figure 3.

#### 2.2.2. Implementation

In a previous study on ZigBee devices for heat measurements in fire, we observed significant delays for the data gained on the WSN when compared with that for equivalent wired devices [6]. We used an OCARI protocol stack to offer an appropriate quality of service to develop a fire metrology system for the field scale to overcome delays. The aim of this study involves determining if the two innovative elements of the OCARI stack (i.e., MaCARI protocol and its OSERENA scheduling algorithm) when compared with that of the Zigbee one can provide the quality of service if used conjointly for fire measurements. As a consequence of new skills (i.e., MACARI and OSERENA used conjointly), OCARI makes it possible to switch off the network nodes during a cyclic schedule, allowing a wake-up cycle, measurement cycle, and data transmission cycle. Finally, the energy consumption approximately corresponds to 0.3 W per node in the OCARI system relative to 1 W per node for the Zigbee standard as used in Reference [6].

Consequently, the power supply of an OCARI node is actually smaller than a ZigBee one. The Zigbee node works with a 7 Ah lead battery (with size 15.6 cm × 9.8 cm × 6.2 cm and weight 2.2 kg), whereas the OCARI node works with a 0.23 Ah battery (size 4.85 cm × 2.65 cm × 1.75 cm and weight 59 g). Hence, this reduces the size of the entire solution supporting the radio communication node, sensor, and power supply. Finally, the elements of the OCARI network are four times smaller than those of Zigbee. It is advantageous to preserve the node from the fire. Specifically, even if they are buried or protected with ceramic insulation, the surface to bury or cover is evidently smaller with respect to protection. Furthermore, with an adapted topology, if a node breaks out in a fire, the communication can be automatically rerouted by the system without human maintenance. Finally, when a sole node cost is ten times smaller than a wired datalogger, it is easier to replace the same if it is destroyed by a fire.

Finally, a special OCARI board was developed to guarantee the timestamp of each analogue data digitised before the radio message packaging. The new system timestamps data at the instant of their digitisation by the analogue-to-digital converter (ADC) via a real-time clock circuit implemented on the board. Subsequently, the open-source OCARI stack was modified to compound messages, including the instant of the measurement, value of the measurement, and 11 other pieces of networking information (i.e., parent identification number, node identification number, intensity of radio signal, etc.):A four times more compact node architecture that results in a lighter sensor node and enhanced resistance to fire exposure;increased performances in terms of energy consumption (0.3 W-a-node for OCARI versus 1 W-a-node for Zigbee when devices work as a full router);accurate timestamping of the data when compared to a standard wired datalogger solution.

## 3. Experimental Protocol

The present study evaluates the ability of an OCARI-based WSN to break the limitations observed with a Zigbee-based WSN for heat measurements during fire experiments, especially in terms of data timestamping. To the best of the authors’ knowledge, extant studies do not examine experiments involving the use of OCARI as a measurement system, specifically in the case of heat measurements in a real fire. The most similar study was performed by the authors of [6], where strong limitations were noted in the use of Zigbee to replace a wired system. The principle of the evaluation presented here is identical to that in Reference [6] as follows: It consists of comparing data recorded with the new WSN system based on a typical wired solution with the Campbell Scientific CR3000 datalogger. In the following, we first present the devices for heat measurements and wired and wireless solutions to record data as well as the setup of each experiment.

### 3.1. Heat Measurement Devices

Heat data are collected with standard thermocouples and radiative heat fluxmeters (HFMs) that are typically used for fire experiments [1,2,3,22,23,24]. We performed two types of heat measurements as follows:*Measurement of gas temperature* (in both air and fuel beds): Gas temperature *T* is measured around the transducer and makes it possible to detect the presence of smoke (60 ${}^{\circ }$C < T < 500 ${}^{\circ }$C) or flame (500 ${}^{\circ }$C < T ) as per fire safety engineering standards;*Measurement of radiant heat fluxes*: The system measures heat radiation from the fire to the target sensor.

The measurement of the gas phase temperature is performed with a K-type thermocouple that corresponds to a transducer that obeys the thermoelectric effect or Seebeck effect. This effect is due to a junction (hot weld) of two electrical wires wherein each is composed of a different alloy at $T_{1}$ temperature. A second junction (cold weld) is at $T_{2}$ reference temperature. The temperature difference between the hot and cold junctions is converted into an electric potential difference or voltage in which temperature exhibits a linear behaviour.

Temperatures are measured using chromel–alumel type thermocouples (K-type) that are sensitive in the $-100{\;}^{\circ }$C to 1370 ${}^{\circ }$C measurement range. The thermocouples exhibit a 250 μm diameter gained junction with a response time of 0.3 s. They are connected to an amplifier module (+/−40 mV for voltage output of the transducer) and allow a full-range temperature measurement capable of facing a large-scale fire of vegetal fuel.

The radiant heat flux measurement is performed via Medtherm (c) heat fluxmeters (HFM). The HFM relies on temperature measurements on the hot and cold sides of a cylindrical shape solid where the hot side is exposed to the heat flux. The incident radiant heat flux is obtained from the measured heat flow passing through the solid on the exposed side. The heat flux sensor calibration lies primarily with an exposure to a reference flux emitted by a black body at a high temperature and the radiative source based upon Planck’s radiation law. In the present experiment, we used two models of HFMs: The first obtained 0–20 kW/m${}^{2}$ as the maximum level of heat radiation, and the second obtained 0–200 kW/m${}^{2}$.

All Medtherm heat fluxmeters are manufactured with a measurement error corresponding to 3% of the responsivity, with a coverage factor corresponding to 2 for a 95% confidence interval based on the Medtherm black body calibration report for two measuring ranges (0–20) (Schmidt–Boetler gauge) and 0–200 kW/m${}^{2}$ (Gardon gauge). For example, a domestic oven emits approximately 5 kW/m${}^{2}$ at full power. The Medtherm HFMs produce a signal between 0 and 10 mV, and this is in accordance with the analogue inputs of the OCARI’s data acquisition cards constructed for the OCARI’s nodes. Calibration is set according to the ISO9014 standard for heat measurements in fire engineering [25].

This difference in HFM sensitivity is noted as LHFL for the 0–20 kW/m${}^{2}$ range (low-heat flux level) and HHFL for the 0–200 kW/m${}^{2}$ range (high heat flux level). With respect to a given fire in the open, the HFMs typically present a slight difference when plugged on the standard wired datalogger used here. The study involves examining the effect of the difference in sensitivity when used with the OCARI wireless system.

### 3.2. Data Acquisition Systems

#### 3.2.1. Wired System: Campbell CR3000 Datalogger

On the wired network, the heat quantities—namely, temperature and radiant heat flux—are recorded using a portable CR3000 datalogger model (Campbell Scientific—Edmonton, AB, Canada ©). This is dedicated to work in outdoor conditions and was already used to collect data during large-scale fire spreading experiments [2,6,22,23]. It also serves as the reference wired datalogger to evaluate the response of the Zigbee WSN tested in Reference [6]. For a fair comparison, we used the same wired datalogger with an identical setup. Specifically, on the CR3000, there are 14 differential analogue channel inputs. Each channel exhibits a maximum scan rate of 100 Hz. The recorded voltage levels are digitised with a 16-bit resolution. We selected five voltage levels ranging from ±50 mV to ±5 V. In this study, the channels are configured at a ±50 mV (analogue resolution: 0.67 μV per digit). We selected a sampling rate corresponding to 1 Hz. The data were stored in 4 MB of memory. It is necessary to bury the wires from the devices to the datalogger into the ground to protect them from the effects of the fire. Furthermore, a thermal insulation composed of an assembly of ceramic fibre protects the off-ground section of the datalogger. This protection is intended to make the wires resistant to the fire. The analogue data were encoded on 16 bits on the wired datalogger and timestamped at the instance of the digitisation. We estimated the delay for transmitting the information from the transducer to the datalogger for a cable with a length ranging to a few metres as approximately 9 ns based on the CR3000 datasheet. We can therefore neglect the corresponding delay in transporting the signal from the sensor to the ADC, in the wired architecture at least, with the chosen 1 Hz sampling rate.

#### 3.2.2. OCARI-Based Network for Heat Measurements in Fire Experiments

The wireless platform runs on the OCARI software prototype for wireless communication based on the Adwave ADWRF24-LRS technology (Figure 4). It generates a digitised data flow from the measurement area to the network coordinator (CPAN).

Each OCARI network node consists of a radio communication device based on an Atmel microcontroller SAM3. Each radio device is coupled with an acquisition interface in which two analogue transducers are connected through a 12-bit ADC on which a real-time circuit (RCT) operates. The RCT is integrated to generate a timestamp at the moment of analogue-to-digital conversion. This type of a timer circuit device is energy-consuming, although the ability of OCARI stack to promote very low energy consumption solutions increases due to reasonable RTCs. In the active mode, the RTC circuit uses 0.0825 W, whereas consumption decreases from three orders of magnitude in the power-down period of each cycle. This leads to an overall cost of an OCARI solution, which scales 0.4 W as the maximum energy consumption for the expected and extremely accurate timestamping of the data. The first input reads the signal from heat fluxmeters, whereas the second input is plugged into the range from the thermocouple (Figure 5).

The wireless platform is therefore based on the OCARI 2015 release and was developed in cooperation with Adwave. It simultaneously records air temperature and radiative flux in a fire. A few adjustments were made on the nodes connected to the measurement interfaces: The analogue heat sensors must be conditioned for being plugged into OCARI hardware. For the sake of readability, details of the conditioning are reported in Appendix B.

### 3.3. Experimental Plot

In configurations 1 and 2 (Figure 6 and Figure 7), we conducted three large-scale fire experiments. Specifically, 10 m${}^{2}$ of fuel bed was formed by spreading excelsior with a 2 to 8 kg/m${}^{2}$ fuel load from a height ranging from 50 to 90 cm, and this resulted in flames with heights ranging from 1 to 4 m. Facing the flame front, a set of transducers is displayed by a couple (i.e., two thermocouples or two radiative heat fluxes), and each element of a couple is plugged on a wireless node or its wired counterpart, namely the CR 3000 Campbell Scientific. The main experiments characteristics are reported in Table 1. The horizontal wind speeds are recorded using a 2D ultrasonic anemometer sampled at 1 Hz.

Two configurations were selected to monitor the fire, namely a vertical monitoring of the horizontal component of heat radiation (i.e., parallel-to-the-ground, C1, Figure 6) and horizontal monitoring of the same horizontal component in heat radiation (C2, Figure 7). The specifications of measurement devices used with nodes are summarised in Table 2 for configuration 1 and Table 3 for configuration 2.

Configuration 1 differs from configuration 2 with respect to the position of measurement points in the axis of the plot and the nature of the measurement points. Specifically, the aim of C1 involves displaying central HMFs in a vertical direction, whereas C2 sets them in a horizontal direction. In the C1 configuration (Figure 6), sensors are located at a height of 0.5 m above the ground for nodes 2, 3, and 4, while node 1 is at a height of 1.5 m from the soil. Specifically, C1 evaluates the heat radiation that impacts a vertical axis at a distance of 3.3 m from the left edge of the plot (Figure 6). Additionally, C2 investigates the penetration of the heat radiation along the central axis of the plot (Figure 7), and each sensor is located at a height of 0.5 m. Sensors 2 and 4 are separated by a distance of 1 m in the C2 configuration. Furthermore, another difference exists in temperature measurements between both configurations. Specifically, C1 leads to a single temperature measurement at the centre of the vegetal plot (N5 in C1), while C2 leads to two temperature measurements along the central axis of the plot (N2 and N4 in C2). Each node is identified by an integer corresponding to 1, 2, 3, or 4 on the experimental WSN, as reported in Table 2 (respectively, Table 3) for the C1 configuration (respectively, C2). The wired and wireless thermocouples are separated by a distance of less than 0.5 cm, thereby allowing the assumption that they measure the same local temperature.

The aim of the present study also involves observing the effects of HFM sensitivity when implemented on the OCARI-based WSN. Thus, two sensitivities of the HFM were used for the comparison of the wireless system with the wired system as follows: 20 kWm${}^{2}$ (N3 and N4) and 200 kWm${}^{2}$ for embedded measurement points (N5 and N7).

In configuration 2, nodes N1 and N2 are connected to 20 kWm${}^{2}$ HFMs (referring to LHFL data on plots), whereas nodes N3 and N4 are plugged with 200 kWm${}^{2}$ (referring to HHFL data on plots). Please note that these experiments did not involve a large spanned wireless network, which is the aim of an operational WSN. Hence, we did not investigate the network failures as a communication channel occupation and possible congestion due to several nodes [26]. We focused on the ability of the WSN to conveniently record heat data and the relevant timestamped data when compared to a standard wired system. The fire is ignited from a fire line on the downwind edge of the plot using a constant quantity of alcohol and a fire torch.

It is important to consider that the experimental configuration leads to variations in the distance in terms of the height from fire to sensors, load, and wind conditions. The aim involves promoting a large set of analogue signals from the WSN and the wired datalogger to compare the same and does not involve investigating different incident radiations from fire to the sensing target. The results are presented in the next section and focus on the influence of measurement conditions and sensor sensitivity on signals gained by WSN when compared with the ones gained on the wired datalogger. The physics of the heat radiation in outdoor fires is not discussed in the present study. Readers can refer to publications focusing on this subject [2,3,23] for details.

Another important point concerns the thermal protection of each node and their analogue sensor: each node is inserted into a shield pocket made with ceramic wool. Into their thermal shield of ceramic wool, the sensor nodes—i.e., the transducer and the radio-communication node—can show resistance to a distant heat irradiance as to a prolonged contact with the fire. However, if the shape and size of the shield of the sensor nodes can be set up according to an incident heat radiation, this is not the case when these sensor nodes enter into contact with the fire. The direct contact between fire and target promotes a mixed heat environment including radiative and convective contributions of heat transport from flames to the node. If the irradiation is well known (the heat flux sensors were calibrated against a black body source in the same conditions), this is not the case for the convective heat transfer. Typically, if a fire causes irreversible damages to the shielded nodes embedded, we would not be able to quantify the level of heat flux responsible of these damages because the convective heat flux cannot be measured (since heat flux sensors are calibrated against a radiant source). For instance, in Reference [6], we observed that a fire across a 10 kg/m${}^{2}$ load of vegetal fuel causes the destruction of nodes. However, we could not quantify the amount of heat causing this destruction because of the uncertain contribution to the heat convection. This is why we do not embed sensor nodes into the fire in the present study as we did in Reference [6], to remain under the influence of a sole heat radiation for which shields are dimensioned and avoiding the sensor destruction.

## 4. Results

In our experiments, we compared signals gained by a wired datalogger and OCARI-based WSN in the same experimental conditions. Figure 8, Figure 9, Figure 10 and Figure 11 show comparisons between wired (continuous lines) and wireless signals (symbol lines). In each figure, there is a temperature plot (top) and a heat flux plot to compare between LHFL and HHFL (bottom) in configurations 1 and 2, for 2, 4, and 8 kg/m${}^{2}$.

As shown in configuration 1, there are fires that display behaviour that is consistent with the usual observations as follows: The level of heat radiation is measured as 3.3 m in front of the edge of the plot and increases with the fuel load under low wind conditions (Table 1). In configuration 1, the heat flux density reaches 8, 10, and 20 kW/m${}^{2}$ with respect to 2, 4, and 8 kg/m${}^{2}$ (Figure 8, Figure 9 and Figure 10, respectively).

As shown in configuration 2, the wind flow can affect the expected behaviour of the fire observed at a field scale under low wind conditions. For example, the 8 kg/m${}^{2}$ fuel load (Experiment C2 3 in Table 1) does not coincide with the stronger heat radiation because the wind is not conveniently oriented to sustain the development of a vertical flame front as the source of the heat radiation, which faces the HFMs, as shown in configuration 1. In the present case (i.e., 8 kg/m${}^{2}$, configuration 2, Figure 11, the flame front is tilted in a transverse direction, and this reduces the view factor between the flame front and HFMs (see photograph in Figure 7). Thus, the horizontal component of heat radiation is lower than that in the case of 8 kg/m${}^{2}$ in the first configuration (approximately 20 kW/m${}^{2}$ in Figure 10). The influence of the unexpected local instantaneous aerology is valuable to test the wireless heat measurement system in outdoor conditions.

We now compare the time evolutions of signals obtained with both wired and wireless solutions facing the fires. The first main difference between wired and wireless solutions involves the temporal resolution of the rapid increase and decrease in temperature and flux plots. For example, we consider the temperature profiles in configuration 1 at the top of Figure 8 for the 2 kg/m${}^{2}$ fuel load, Figure 9 for the 4 kg/m${}^{2}$, and Figure 10 for 8 kg/m${}^{2}$. All plots exhibit a different time sampling between both wired and wireless data acquisitions with similar temperature signals. Specifically, the wired datalogger records digitised information from each thermocouple (TC) as sampled at 1 Hz. The wireless system works differently wherein it records data packaged in each message reaching the coordinator of the personal area network (CPAN) successfully. In each message, timestamping of the AD conversion exists at every note at the microcontroller level (Figure 5). Therefore, in each figure from Figure 8, Figure 9, Figure 10 and Figure 11, we observe data and the timestamp that was received by the CPAN from the WSN. This implies that the sampling of analogue signals on the WSN is observed a posteriori as the reconstruction of the couple timestamp-data in each validated message. Given the possible constant message losses in this type of a harsh environment (e.g., large fires), the WSN data rebuilt by the CPAN can correspond to nonregularly sampled data.

Under the conditions, the OCARI based WSN sends–receives messages at a maximal rate of approximately 0.2 Hz. Therefore, it is not possible to resolve wireless temperature and heat flux signals with the same temporal accuracy as the wired ones. The phenomenon was already observed in Reference [6] with the Zigbee radio communication protocol, which was slightly faster than that of the OCARI protocol (the Zigbee send–receive frequency rate approaches 0.5 Hz). However, the OCARI protocol introduces relevant timestamping of the data selected as the instant of the AD conversion despite a lower a priori time resolution than Zigbee. This is because all microcontrollers share the same absolute time base. This technological upgrade of the wireless system is possible with the OCARI stack and eliminates delays from WSN to wired signals observed with Zigbee in Reference [6]. Specifically, in Zigbee solutions, delays in data reporting occurred when messages are delayed or lost with the standard timestamping of the data, i.e., with the date on which messages were received at the central node. This results in the time shift of the wireless signals from the wired ones that reached up to 10 s in the largest fire experiments at a real scale [6] (i.e., in a 4-m high and 5-m long flame front). Irrespective of the configuration, Figure 8, Figure 9, Figure 10 and Figure 11 easily observed that the wireless signal is no longer time-shifted from the wired one irrespective of the case when the OCARI solution is used. Finally, OCARI technology overcomes the technological Zigbee limitations to record heat data in natural fire experiments.

Subsequently, we present the temperature gained at the centre of the vegetal plot in the C2 configuration and for each fuel load (i.e., from Figure 11 at the top to Figure 12). We focused on configuration 2 because two temperature measurements were conducted on the vegetal field bed (Figure 7). In this case, each record on central nodes 2 and 4 follows the same temporal evolution. The temperature initially increases at node 2 and then at node 4 because the fire ignited on the left side of the vegetal field bed spreads to the right side, and thus reaches node-2 TC. On each of the central nodes 2 and 4, from its value in the ambient air, the temperature suddenly increases when the flame front reaches the sensor, fluctuates at approximately 800 ${}^{\circ }$C, and then decreases more or less quickly (Figure 12 for 2 kg/m${}^{2}$, Figure 13 for 4 kg/m${}^{2}$, Figure 11 at the 8 kg/m${}^{2}$ top in configuration 2). Similar behaviour is observed in configuration C1. Finally, in each case, irrespective of the configuration of the wireless system, it follows the wired reference signal. However, beyond the aforementioned sampling difference between signals, we observed that the signals do not exhibit an exact match in terms of amplitude. Specifically, this happens during the steep increase in temperature above 500 ${}^{\circ }$C, thereby indicating that the contact of the thermocouple with the flame front and local maximum/minimum peaks of temperature on the WSN do not coincide with the wired ones (i.e., the time interval [40–60 s] in Figure 8 for the 2 kg/m${}^{2}$ in configuration 1). The difference is initially due to the thermocouple sensitivity to the local radiation or convection gain/loss. It typically scales to an error of approximately 10% in temperature measurements [6]. However, the difference in amplitude from the wired one to WSN can also exceed the range of 10%. For example, in configuration 1 at 8 kg/m${}^{2}$, the error reaches up to more than 15% on the time interval [13;144] s. This indicates that beyond the TC error, another effect occurs and is related to the WSN system to explain the difference in the signal amplitude for temperature measurements performed at the same location. This effect is necessary due to the ADC difference in resolution between wired-(16 bits architecture) and wireless-(12-bits architecture). Both sources of errors, namely radiation or convection gain/loss of the TC and the difference in bit resolution, participate in explaining the 15% discrepancy in amplitude observed between the WSN and wired TC measurements in the fuel bed.

We now consider the heat flux measurements. Specifically, the explanations provided to describe the differences in temperature signals remain valid for those observed in the heat flux measurements. The comparison of heat flux measurements (Figure 8, Figure 9, Figure 10 and Figure 11 bottom) also explores a few other aspects while comparing the wired datalogger and WSN. We consider the impacts of HFM sensitivities to low-level heat flux (LLHF) and high-level heat flux (HLHF) on the signals gained with wired or wireless network technologies. To the best of the authors’ knowledge, the aforementioned point was not explored in existing studies, including those with the Zigbee prototype.

Specifically, we consider the impact of the heat radiation on vertical targets that can be observed on the HFM plots in Figure 8 for 2 kg/m${}^{2}$, Figure 9 for 4 kg/m${}^{2}$, and Figure 10 for 8 kg/m${}^{2}$ in configuration C1. A couple of HFMs with the same sensitivities are represented with the same colour between wired (continuous lines) and wireless (symbol lines) systems.

In configuration 1, with respect to the 2 kg/m${}^{2}$ cases (Figure 8), low-level HFMs on the WSN (node 2 (red)/central top and node 3 (blue)/lateral) collapse in a single master curve with LLHF meters plugged on the wired datalogger. Departures from wired to wireless signals approximately correspond to 0.4 kW/m${}^{2}$, i.e., lower than each confidence interval and scale the maximal measurement error of the radiant heat flux due to the heat flux Medtherm transducers. This difference from wired to wireless HFMs increases to 3 kW/m${}^{2}$ in the 4 kg/m${}^{2}$ case/configuration 1 (Figure 9) and reaches more than 2.5 kW/m${}^{2}$ in the 8 kg/m${}^{2}$ case (Figure 10). Thus, discrepancies are stronger between wired and wireless HFMs that are calibrated for a high level of radiant heat fluxes (green (node 1—lateral) and black plots (node 1—central bottom). This begins from 0.43 kW/m${}^{2}$ at 2 kg/m${}^{2}$ and extends to more than 10 kW/m${}^{2}$ in the 8 kg/m${}^{2}$.

This behaviour is confirmed in configuration C2, where the horizontal penetration of the heat radiation is investigated. Three fuel loads are examined in Figure 12 for 2 kg/m${}^{2}$, Figure 13 for 4 kg/m${}^{2}$, and Figure 11 for 8 kg/m${}^{2}$ in C2. In this case, the profiles of LHFLs are black (node 1 in C2—lateral) and red coloured (node 2 in C2—front central), whereas HHFLs are blue (node 3—lateral) and green (node 4—back central).

Thus, the difference between wired and wireless heat flux data is stronger than the calibration errors of the fluxmeters. Therefore, this is caused by the bit-resolution effect of each system: The wired one corresponds to a 16-bit architecture, while the wireless one only offers a 12-bits ADC on the data acquisition board. The second display of the bit-resolution effect indicates that an upgrade of the wireless systems towards the accuracy of the wired one requires a higher bit resolution of the ADC component on each wireless node. Specifically, the upgrade must be on purpose, keeping in mind the requirement to maintain a reasonably low cost of the WSN solution to remain consistent with the advantages of the microcontroller-based wireless technology.

Finally, the proposed OCARI-WSN system fully solves the time-shift effect on temperature and heat flux signals as observed with the previous Zigbee-based measurement solution. The temporal plot of the WSN follows the relevant wired reference signal. The temperature and heat fluxes acquired on the WSN suffer from a difference in amplitude when compared to that acquired via the wired one. This is due to the measurement error relative to each sensor used and also due to the difference in bit resolution between both systems.

We discuss the quality of the results in the following section to evaluate the opportunity to design an industrial OCARI-based solution to monitor fire scenarios in natural and confined conditions.

## 5. Discussion

Given a strategy of recording the data over an extended area where a catastrophic fire is possible, the wireless solution exhibits a priori low-cost and low-energy consumption when compared to the wired one. Although previous studies illustrated that electronics for the data acquisition system can be embedded into a real-scale fire ([2,3,24] for wired systems and [6,27] for wireless systems), the protection of each node of a wireless system from the fire is evidently cheaper and easier than each element of a wired one. Wired systems are more vulnerable because the heat propagates along the wires from the transducers immersed in the fire up to the data acquisition system. Therefore, the entire length of wires must be thermally isolated. Furthermore, in both cases, the systems must guarantee its own innocuity, i.e., it works without causing any hazard itself. Thus, the entire system works to prevent damage due to its own operation, especially in an explosive atmosphere (as liquid and gaseous fuel storage areas).

With respect to the previous results, the following points can be made irrespective of the concerned configuration:The time shift is solved by the timestamping allowed by the OCARI stack;the measurement of the temperature by the wireless system matches that of the wired system with respect to the limit of the errors, both due to the sensitivity difference between transducers and ADC difference between the wired and wireless architectures;the measurement of the heat flux by the wireless system introduces spurious behaviours of the signal when compared with the wired system in the case of high heat flux transducers.

These observations lead us to discuss the key feature of the OCARI-WSN. It is an open-source generic solution that can be easily adapted to each sensor available in the industry. Specifically, the temperature measurement is acceptable for fire detection and monitoring because the relative error (10–15%) is poorly dependent on the solution. The amplifier for the output signal transducer on the OCARI device is well-adapted to the temperature measurement requirements with thermocouples. Switching off the cold-junction voltage to increase the bit resolution (Appendix B) makes the OCARI system stable during the measurement with a TC signal covering the full digital range. Despite both the measurement error and lower frequency sampling in the OCARI solution, the recording of voltage from the TC transducer is acceptable on the wireless system with a lower energy consumption and a lower cost when compared with those of Zigbee.

In addition to the resolution of time delays observed with Zibgee and energy consumption reduced from a factor 3, OCARI leads to smaller nodes. Indeed, for an equivalent duration of fire experiments (about 300 s) presented in Reference [6] as in that study, the enhanced performances in energy consumption can be illustrated by the size reduction of the load cells: A 6 Ah Pb load cell-a-node is needed for the Zigbee full function device protocol (each node is a router), whereas a 1.2 Ah load cell-a-node is sufficient for the OCARI FFD one. The enhancement of the OCARI-based solutions in term of energy consumption and compactness of each node proceeds also from the size reduction of the load cell. This is illustrated in Figure 4. Therefore, each node is more compact, and thus, it is lighter and easier to deploy, maintain, and protect from the fire. Thus, it can potentially be used to measure temperature in the gas phase or on solid industrial infrastructures with a competitive cost and open-source promising features. This offers generality and flexibility to the user.

However, the OCARI performance for heat measurement is not extremely evident. The HFM is a sensor that works closely with a TC. A voltage difference is generated as a function of a temperature difference through the sensor solid body from which the heat flux can be estimated (through a calibration process). We tested here the easiest and cheapest solution for adapting HFM to the OCARI architecture using the signal amplifier designed for temperature measurements. Thus, we plugged the HFM over an instrumentation amplifier dedicated to amplifying a millivolt signal—namely, the [0;10] mV interval up to the [0;3.3] V interval—and switching the temperature compensation off.

The consequences of this strategy are observable and include the following: During a prolonged fire scenario, the HFM signal on OCARI fluctuates under the influence of continuous heat radiation due to a fluctuating voltage reference and a lower bit resolution than the required one. Finally, the sensor does not deliver an accurate heat flux measurement through the OCARI solution anymore and the effect becomes unacceptable when high density heat fluxmeters are used.

The adaptability of a new transducer to OCARI requires the development of amplifiers dedicated to the exact sensor range towards the input of the OCARI ADC and, specifically in the case of high-level HFMs (0–200 kW/m${}^{2}$), with a stable reference tension. This is indispensable in terms of supporting an extended range of analogue tools through the OCARI interface with an industrial level of quality. In the conditions, it is expected that every other analogue sensor (gas detection, liquid level, pressure, strain, deformation, or moisture measurement) can be performed in the present timestamped OCARI setup if sampling with a maximum rate of 0.2 Hz is sufficient for the quality of the measurement and if a dedicated solution for amplification is developed as compatible with the rest of OCARI hardware.

## 6. Conclusions

The study presented a solution to overcome the limitations of WSN for heat measurements during outdoor fires and specifically for the ones observed with the Zigbee communication protocol [6]. The advantages of using an OCARI-WSN for this type of applications include cost compression due to wire suppression as the modularity and flexibility of the measurement system in an open-source WSN. During the detailed presentation of the WSN solution, sensors, and standard datalogger, we discussed the different opportunities ranging from cable heat transducers to voltage amplifiers for interfacing sensors with the WSN node. Subsequently, we evaluated the performance of the OCARI-based WSN for temperature and heat flux measurements. We compared the data recorded on the wireless system to the ones gained on a standard wired datalogger during a series of fire experiments in the open. This led to a more compact and lower energy-consuming solution for a given financial cost, and the results exhibited a convenient recording of the temperature signal with the wireless system, despite a significantly lower sampling frequency when compared with that of the wired one. Specifically, the data are conveniently timestamped in the OCARI version of the solution, and this is in contrast to the Zigbee one in Reference [6]. Differences between wired and wireless solutions in temperature signals are due to the error of each transducer and the bit-error and result from the differences in the 12-bit WSN and 16-bit of the wired solution. The recording of radiant heat flux is more subject to caution with sensors calibrated for a high level of heat radiation (up to 200 kW/m${}^{2}$). In this case, a solution of signal amplification must be specially designed to reduce the error in the HFM signal when implemented on the OCARI-based WSN.

Furthermore, when the limitations are overcome, it allows the deployment of the solution over a larger spanned area and involves a larger number of nodes. Thus, it is necessary to investigate the parametrisation of a large network with respect to the time interval for sending/receiving messages, data packet generation rate, and output power level for transmission to scale the possible congestion of the communication channel [26]. Under the aforementioned conditions, the optimisation of the algorithm to reduce delay and energy consumption could constitute a problematic question due to intense traffic near the CPAN (sink) node [28].

At the end, another outlook of this work is the following: The lower energy consumption skill of the OCARI-based solution can probably also serve to break another limitation observed with Zigbee, i.e., performing long-term measurements because of its energy consumption. Let us recall as a conclusion that except Reference [6], the literature does not report on Zigbee-based WSN systems able to perform fire measurements. The list [7,8,9,10,11] refers merely to detection.

## Figures and Tables

**Figure 1 sensors-19-00158-f001:**
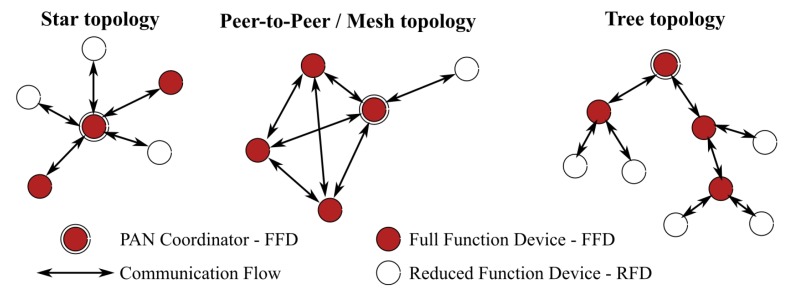
*IEEE 802.15.4* standard networking topologies.

**Figure 2 sensors-19-00158-f002:**
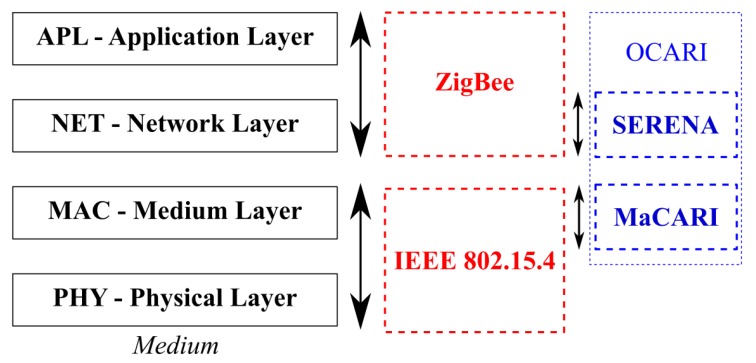
IEEE 802.15.4/ZigBee stack and ‘Open Communication Protocol for Ad Hoc Reliable Industrial Instrumentation’ (OCARI).

**Figure 3 sensors-19-00158-f003:**
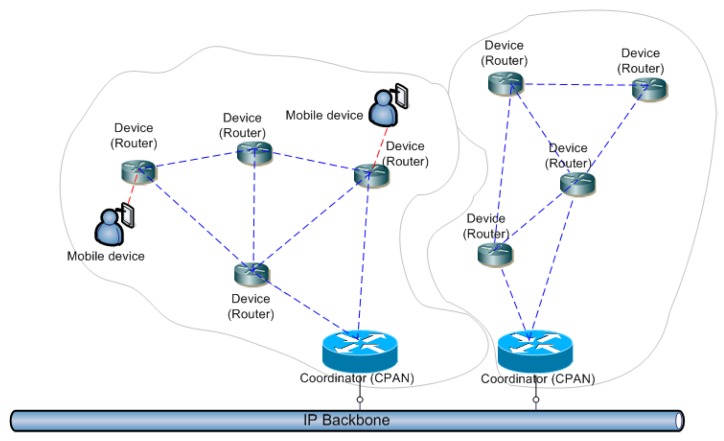
Different nodes of an *OCARI* meshed network.

**Figure 4 sensors-19-00158-f004:**
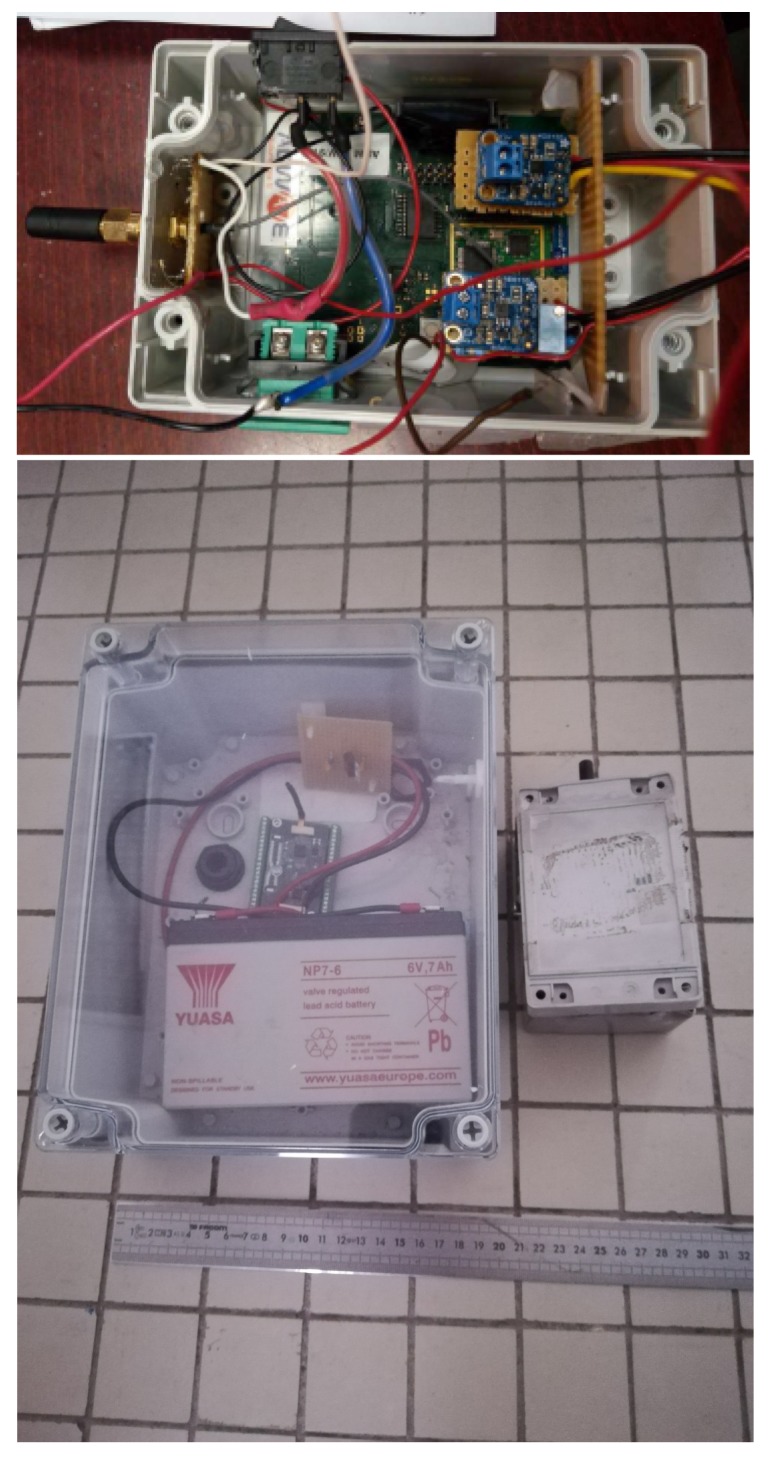
ADWAVE Node (**Top**): Analogue-to-digital converter (ADC) input for the heat fluxmeter (0–10 mV) and thermocouple (0–40 mV). (**Bottom**): Comparison of the Zigbee solution (**left**) and the OCARI solution (**right**).

**Figure 5 sensors-19-00158-f005:**
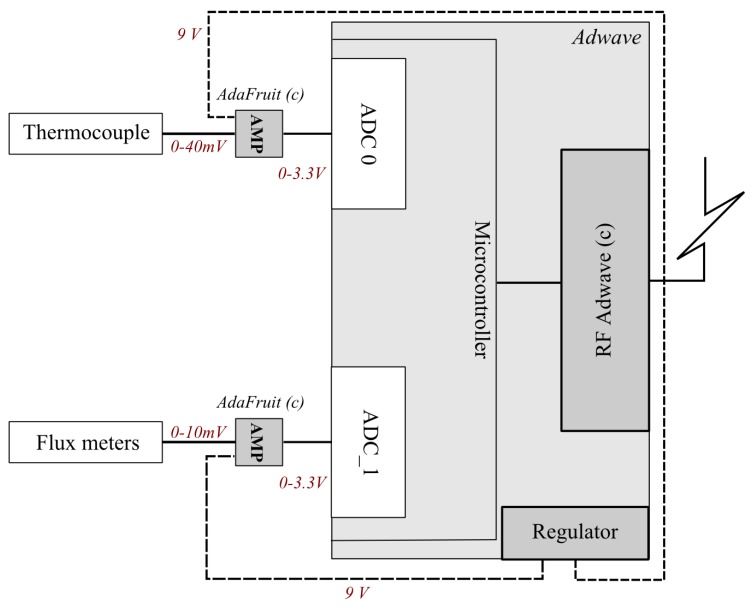
Heat sensor interface for an OCARI node.

**Figure 6 sensors-19-00158-f006:**
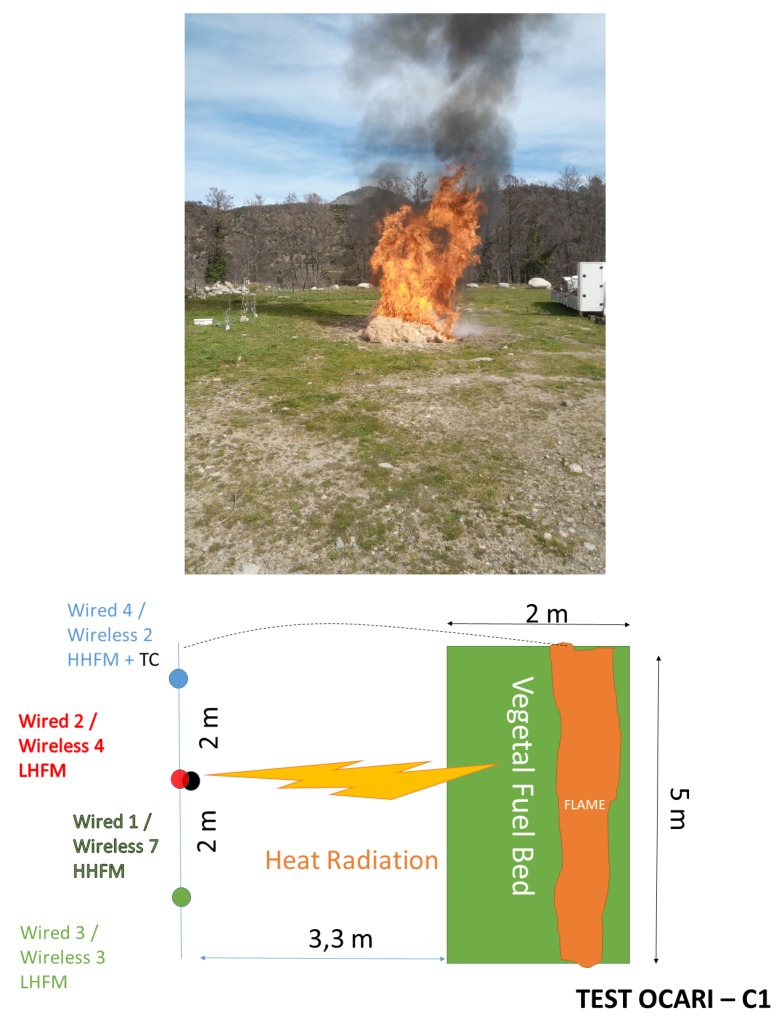
Experimental protocol for large-scale fire test in configuration C1: The 8 kg.m${}^{2}$ case is presented (**top**).

**Figure 7 sensors-19-00158-f007:**
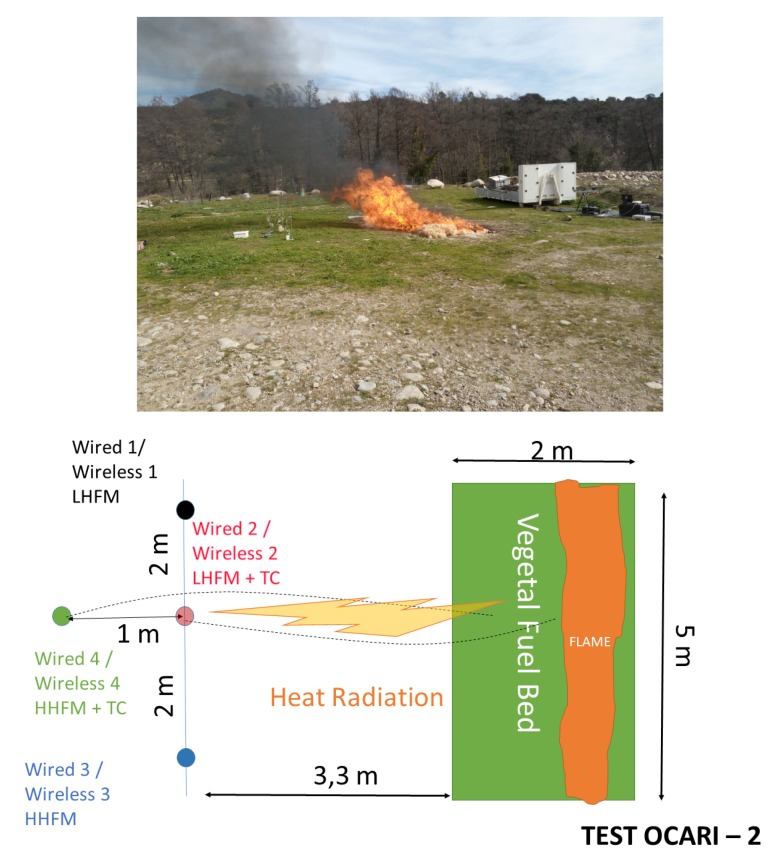
Experimental protocol for large-scale fire test in configuration 2: The 8 kg.m${}^{2}$ case is presented (**top**). Given the wind effect on the fire, the view factor between the flame front and sensing region is lower than in the previous case (configuration 1) and corresponding heat radiation is lower.

**Figure 8 sensors-19-00158-f008:**
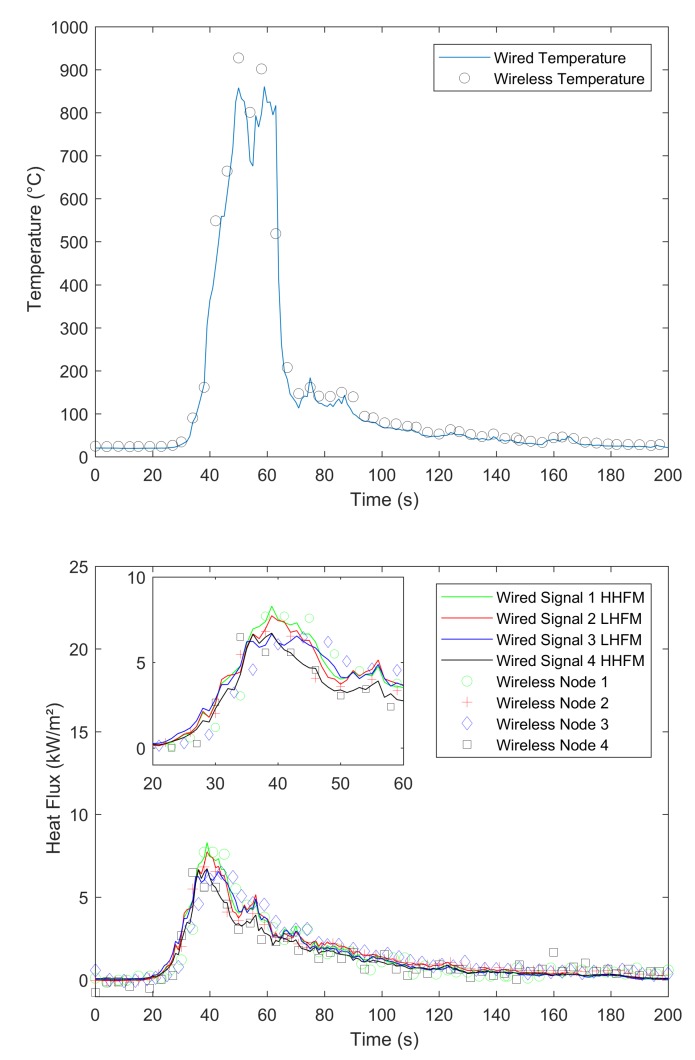
Comparison between the temperature and radiant heat flux acquired on the wired and OCARI wireless system during the 2 kg/m${}^{2}$ fire (configuration 1).

**Figure 9 sensors-19-00158-f009:**
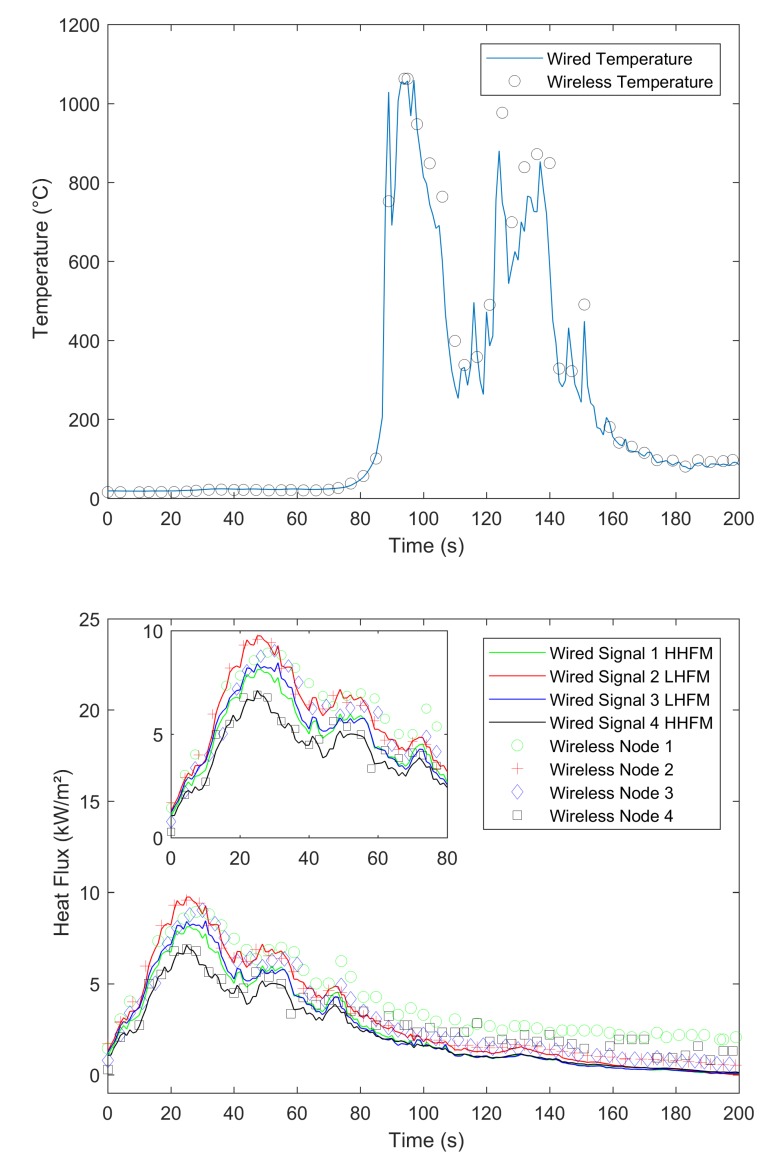
Comparison between the temperature and radiant heat flux acquired on the wired and OCARI wireless system during the 4kg/m${}^{2}$ fire (configuration 1).

**Figure 10 sensors-19-00158-f010:**
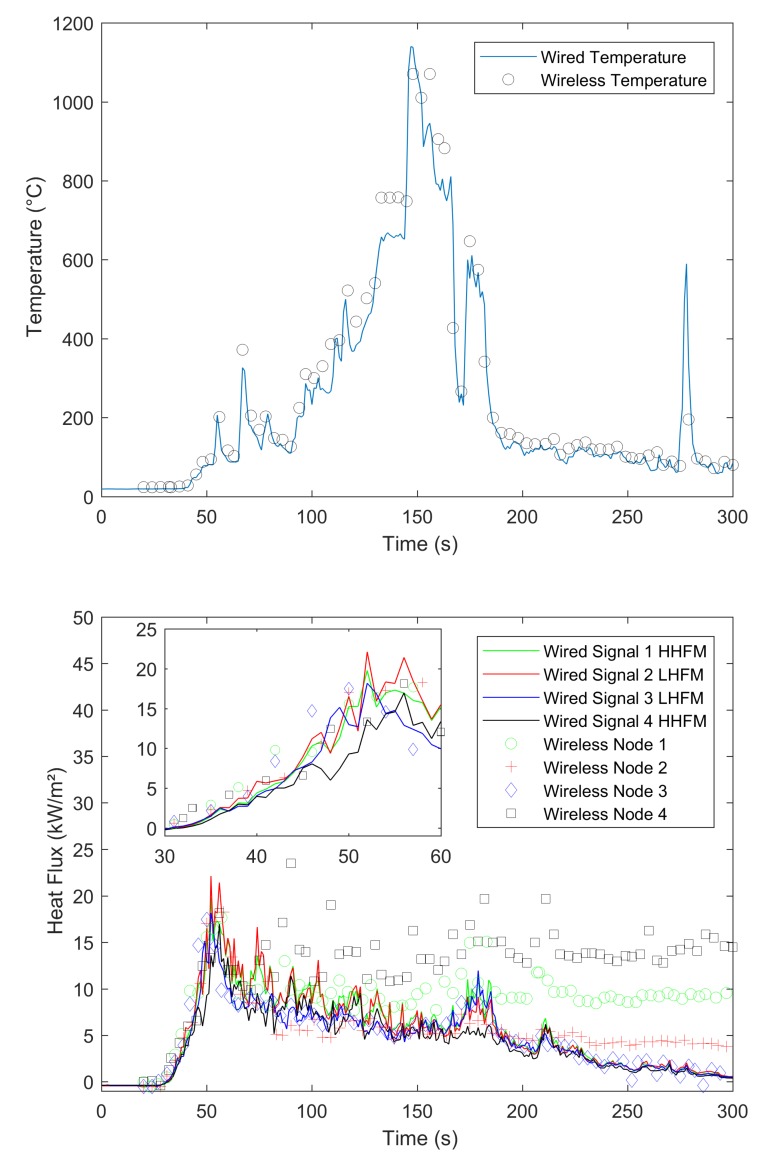
Comparison between the temperature and radiant heat flux acquired on the wired and OCARI wireless system during the 8 kg/m${}^{2}$ fir (configuration 1).

**Figure 11 sensors-19-00158-f011:**
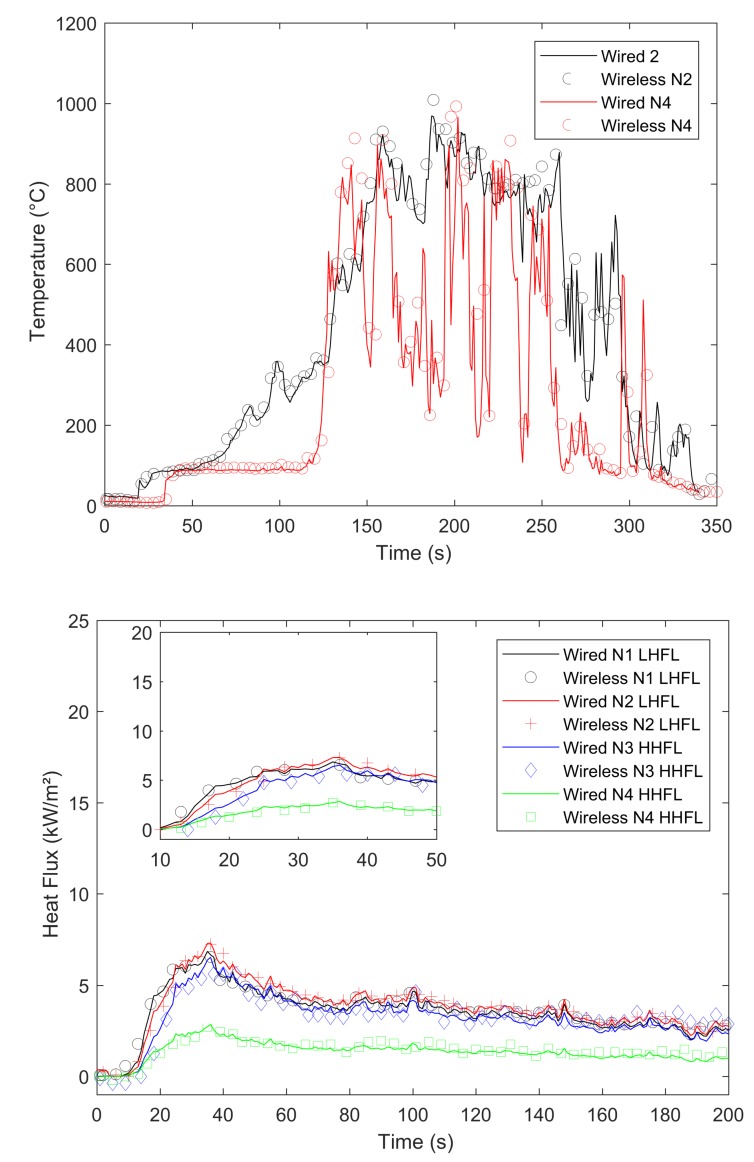
Comparison between the wired *continuous*/wireless *symbols* temperature and radiant heat flux during the 8 kg/m${}^{2}$ fire (configuration 2).

**Figure 12 sensors-19-00158-f012:**
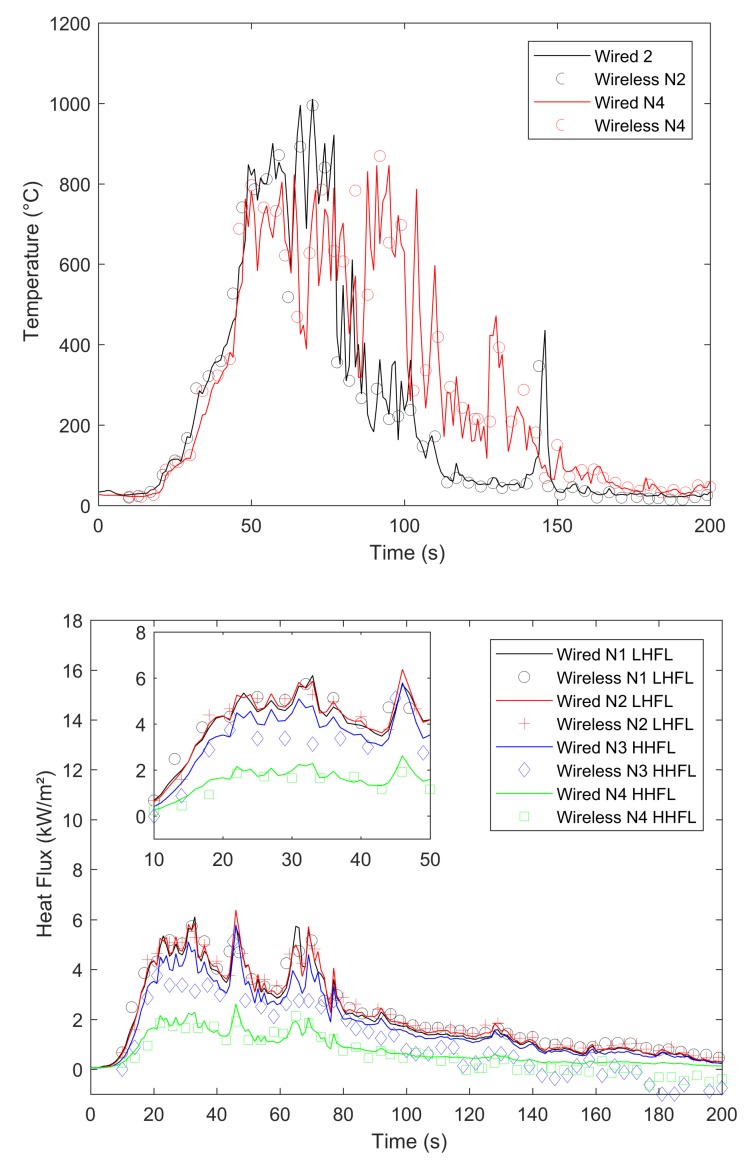
Comparison between the wired *continuous*/wireless *symbols* temperature and radiant heat flux during the 2 kg/m${}^{2}$ fire (configuration 2).

**Figure 13 sensors-19-00158-f013:**
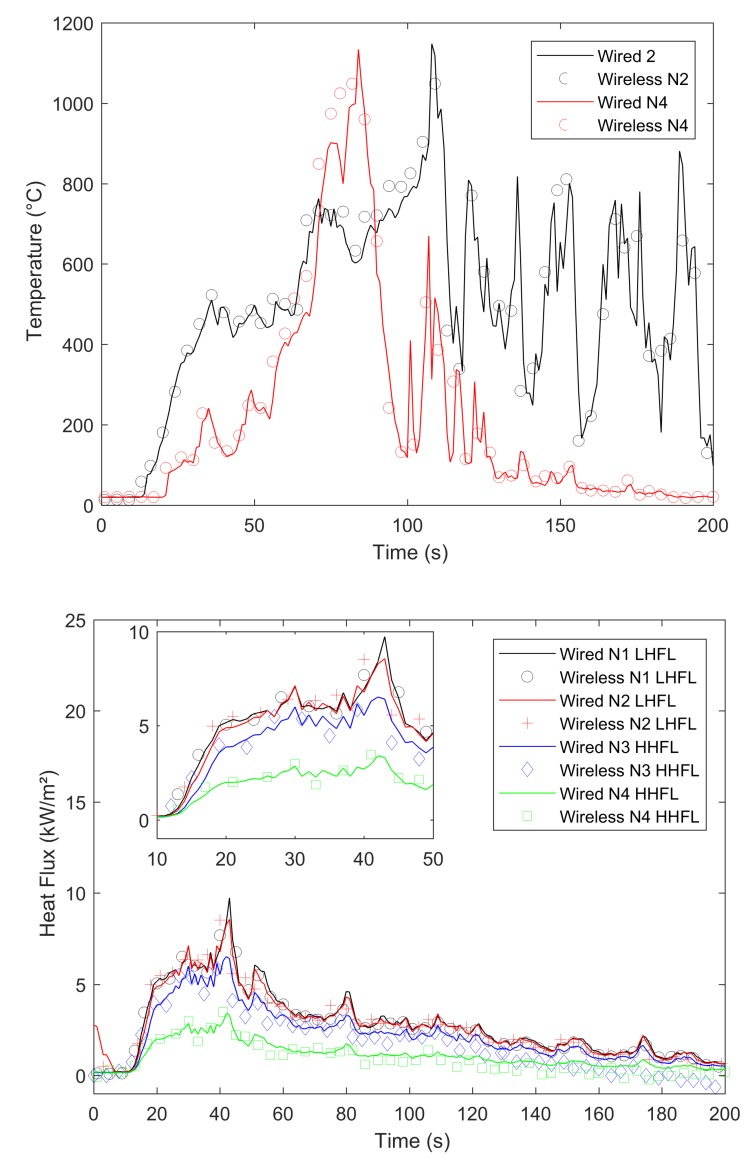
Comparison between the wired *continuous*/wireless *symbols* temperature and radiant heat flux during the 4 kg/m${}^{2}$ fire (configuration 2).

**Table 1 sensors-19-00158-t001:** Large-scale fire experiment settings in configurations C1 and C2.

Experiment	Fuel Load	Wind Direction	Wind Speed
Experiment C1 1	2 kg/m${}^{2}$	Flow opposed to the sensing	3 m.s${}^{-1}$
Experiment C1 2	4 kg/m${}^{2}$	No wind	< 0.5 m.s${}^{-1}$
Experiment C1 3	8 kg/m${}^{2}$	Flow towards the sensing	8 m.s${}^{-1}$
Experiment C2 1	2 kg/m${}^{2}$	Flow towards the sensing area	3.5 m.s${}^{-1}$
Experiment C2 2	4 kg/m${}^{2}$	Flow towards the sensing area	4 m.s${}^{-1}$
Experiment C2 3	8 kg/m${}^{2}$	Flow towards the sensing area	2.4 m.s${}^{-1}$

**Table 2 sensors-19-00158-t002:** Node characteristics in configuration C1.

Node ID	Heat Flux	Temperature
and Position	Maximum Level	Measuring Points
N1 (central at 1.5 m height)	200 kWm${}^{-2}$	Fuel
N2 (central at 0.5 m height)	20 kWm${}^{-2}$	Air
N3 (lateral at 0.5 m height)	20kWm${}^{-2}$	Air
N4 (lateral at 0.5 m)	200 kWm${}^{-2}$	Air

**Table 3 sensors-19-00158-t003:** Node characteristics in configuration C2.

Node ID	Heat Flux	Temperature
and Position	Maximum Level	Measuring Points
N1 (lateral at 0.5 m)	20 kWm${}^{-2}$	Air
N2 (central at 0.5 m, facing the fire)	20 kWm${}^{-2}$	Air
N3 ( lateral at 0.5 m)	200 kWm${}^{-2}$	Fuel
N4 (central at 0.5 m high, 1 m behind N2)	200 kWm${}^{-2}$	Air

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
