# Peer review of "An OCARI-Based Wireless Sensor Network for Heat Measurements during Outdoor Fire Experiments"

_sensors, 2019, doi:10.3390/s19010158_

Reviewer 1 Report

In this paper authors present an implementation of a fire detection systems based on a open source WSN architecture OCARI. They setup a 5 node WSN and prove the effectiveness of the proposed slution on a 10 m2 pine wood. 

This work is very practical, the experimental conditions are limited, as declared by the authors, and their focus was mainly to assess that the proposed solution could be comparable in terms of performance to wired solutions, but being more flexible to deploy. In a previous work authors assessed that a ZigBee solution do not provide same adeauate performance.

The paper is in general  well written,  but the reference description can be improved. In particular, i suggest to authors to createa sub section " related works" where they can give more detail on the ref [7-11] that now are merely listed within the introduction section, and above all describe either the differences of the proposed architecture or the advances with respect to them.

The main wain weakness of this paper is that is self referenced. It seems to be  a step forward with respect to the ZigBee solution provided in ref [6].

Typo: in the OCARI definition both in the abstract that in the introduction there is a ")".

Author Response

Answers to Reviewer 1:

The reviewer’s comments are given in italic font: our point to point answers are given in bold font.

Open Review

English language and style

( ) Extensive editing of English language and style required
( ) Moderate English changes required
(x) English language and style are fine/minor spell check required

The manuscript was entirely revised by Editage English correction services with the advanced editing option.  The corresponding certificate will be uploaded with the revised file.

( ) I don't feel qualified to judge about the English language and style

Yes

Canbe improved

Mustbe improved

Not applicable

Does the introduction provide sufficient background and include all   relevant references?

( )

(x)

( )

( )

Is the research design appropriate?

( )

(x)

( )

( )

Are the methods adequately described?

( )

(x)

( )

( )

Are the results clearly presented?

(x)

( )

( )

( )

Are the conclusions supported by the results?

(x)

( )

( )

( )

Comments and Suggestions for Authors

In this paper authors present an implementation of a fire detection systems based on a open source WSN architecture OCARI. They setup a 5 node WSN and prove the effectiveness of the proposed slution on a 10 m2 pine wood. 

This work is very practical, the experimental conditions are limited, as declared by the authors, and their focus was mainly to assess that the proposed solution could be comparable in terms of performance to wired solutions, but being more flexible to deploy. In a previous work authors assessed that a ZigBee solution do not provide same adeauate performance.

The paper is in general  well written,  but the reference description can be improved. In particular, i suggest to authors to createa sub section " related works" where they can give more detail on the ref [7-11] that now are merely listed within the introduction section, and above all describe either the differences of the proposed architecture or the advances with respect to them.

The reviewer is right: the paper suffers from a concise form about worksrelated with. We corrected the paper in the sense of this recommendation by introducing comments (in red font all along the paper and mainly in the new subsubsection “Related works”) dedicated to present these previous contributions and exhibiting the originality of our approach. The main assertion about this reference list [7-11] is that these studies only use a WSN for fire detection. To our knowledge, there is no experimental study using a WSN for continuous measurements in fire conditions, except [6], with the timestamping limitation described in.

The main wain weakness of this paper is that is self referenced. It seems to be  a step forward with respect to the ZigBee solution provided in ref [6].

The reviewer is right: the bibliography in the paper sounds like only our previous works can be cited. The reason is that our contribution investigated the ability to replace wired data logging systems with wireless ones for measuring physical quantities during a fire (no only detecting the presence of the fire). A first attempt was with Zigbee, the second and present one uses OCARI for a better timestamping. All other studies available in the literature concern fire detections not measurements. The ambition of this work is yet monitoring a fire scenario by continuous measurements for its mitigation. This ambition goes beyond the frontiers of simple binary (no fire/fire) detection.

That excludes to consider that fire detection based on WSN as a relevant measurement system. A detection is not a continuous measurement. The network requirement are not the same.

Indeed, for detecting a fire, you have only to set the wsn-MC based solution on a single send-receive process: a set of nodes is connected to a fire detector (temperature, humidity) and each detection node sends a message to the CPAN when the fire is , for instance,  detected as a high level of temperature or a low level of humidity. In this case, you do not need to transmit each measurement to the CPAN but a single alert is sufficient if detection levels are reached.

Usually, this kind of network just requires an end-device configuration for working: the sensor node does not need to support the continuous routing of the digital information to its neighbours: this skill for which OCARI was developed is extremely energy consuming.  For instance, with standard 2300 mAh load cells, a Zigbee solution does not work more than 24 h in route mode. It only works as an end device, i.e. sending an information to the CPAN only. In an industrial framework, for incstance, Schneider Electrics does not purpose any solution for a continuous routing of measurement flow but point to point message when a 0/1 sensor is activated.

The originality of our proposal in [6] with Zigbee as in the present study with OCARI is to offer a continuous measurement wireless system in which each node is able to measure and to route the digital information through the network.

Please find below further references about this aspect: the WSN remains used as a detection system: no attempt to perform a measurement comparable to a standard wired one exists.

Construction of Wireless Fire Alarm System Based on ZigBee Technology by MA Shu-guang in Procedia Engineering https://doi.org/10.1016/j.proeng.2011.04.662

Wireless Sensor Networks and Fusion Information Methods for Forest Fire Detection, Arnoldo Díaz-Ramíreza, Luis A.Tafoyaa Jorge, A.Atempaa, PedroMejía-Alvarezb in Procedia Technology, https://doi.org/10.1016/j.protcy.2012.03.008

Kechar Bouabdellah, Houache Noureddine, Sekhri Larbi, Using Wireless Sensor Networks for Reliable Forest Fires Detection, Procedia Computer Science, Volume 19, 2013, Pages 794-801, ISSN 1877-0509, https://doi.org/10.1016/j.procs.2013.06.104.

Typo: in the OCARI definition both in the abstract that in the introduction there is a ")".

This is corrected in the revised version of the manuscript.

Reviewer 2 Report

This paper proposed a wireless measurement system for outdoor fire experiments based on an OCARI-based wireless sensor network. Especially the data timestamping problem that previous wireless system has not addressed is overcome in this paper. Following issues could be addressed to improve the paper:

As line 20 expressed, transducers can be embedded in the fire or just near the flame. In figure 6 and 7, table 2 and 3, it seems that the proposed scheme experiment in the latter way. Is there any difference of measuring heat between the two ways?

In line 489, the word  “OCARO”  should be replaced by “OCARI”.

Author Response

The reviewer’s comments are given in italic font: our point to point answers are given in bold font.

Answers to Reviewer 2:

Open Review

English language and style

( ) Extensive editing of English language and style required
( ) Moderate English changes required
(x) English language and style are fine/minor spell check required

The manuscript was entirely revised by Editage English correction services with the advanced editing option.  The corresponding certificate will be uploaded with the revised file.

( ) I don't feel qualified to judge about the English language and style

Yes

Can be improved

Must be improved

Not applicable

Does the introduction provide sufficient background and include all   relevant references?

(x)

( )

( )

( )

Is the research design appropriate?

(x)

( )

( )

( )

Are the methods adequately described?

(x)

( )

( )

( )

Are the results clearly presented?

( )

(x)

( )

( )

Are the conclusions supported by the results?

( )

(x)

( )

( )

Comments and Suggestions for Authors

This paper proposed a wireless measurement system for outdoor fire experiments based on an OCARI-based wireless sensor network. Especially the data timestamping problem that previous wireless system has not addressed is overcome in this paper. Following issues could be addressed to improve the paper:

As line 20 expressed, transducers can be embedded in the fire or just near the flame. In figure 6 and 7, table 2 and 3, it seems that the proposed scheme experiment in the latter way. Is there any difference of measuring heat between the two ways?

The reviewer is right: this point must be enlightened. Into their thermal shield of ceramic wool, the sensor nodes –i.e. the transducer + the radio-communication node- can resist to a distant heat irradiance as to a prolonged contact with the fire.

But if the shape and size of the shield of the sensor nodes can be set up according to an incident heat radiation, this is not the case when these sensor nodes enter into contact with the fire. The direct contact between fire and target promotes a mixed heat environment including radiative and contributions of heat transport from flames to the node. If the irradiation is well known (the heat flux sensors were calibrated against a black body source in the same conditions), this is not the case for the convective heat transfer. Typically, if a fire causes irreversible damages to the shielded nodes embedded in, we could not be able to quantify the level of heat flux responsible of these damages because the convective heat flux cannot be measured (since heat flux sensors are calibrated against a radiant source). For instance, in [6], we observed that a fire across a 10 kg/m² load of vegetal fuel causes destruction of the nodes but we could not quantify the amount of heat causing this destruction because of the uncertain contribution to the heat convection.

This is why we do not embed sensor nodes into the fire in the present study as we did in [6], for remaining under the influence of a sole heat radiation, which is conveniently quantifiable.

This comment is included in the revised version of the paper (blue font at line 421).

In line 489, the word  “OCARO”  should be replaced by “OCARI”.

This is corrected in the revised version of the manuscript.

Reviewer 3 Report

The reviewer appreciate the authors' effort put in this manuscript. However, there exist some following major concerns which need to me addressed:

The authors should emphasize more the contributions in the Introduction part of this manuscript by summarizing its contributions in a paragraph. It is not clear whether this manuscript is the first work to monitor outdoor fires based on an OCARI protocol or not.

In Section 3, detailed illustrations of the measurement devices and whole system should be given. Specifically, in Fig. 4, the author should show the dimension and explain the function of each component used in ADWAVE node. Moreover, its connection to themocouple and flex meters should be presented.

The authors should mention and discuss the heat resistance ability of ADWAVE node and the ability of working stability in a fire.

Figs. 6 and 7 should be represented to increase their clarity. In the upper part, the setup experiment system cannot be seen. While in the lower parts, the authors should show the distance between sensor nodes, the wind direction.

It is stated in the Introduction that OCARI-based measurement system provides lower loss of radio messages compared with Zigbee-based measurement system. However, in Section 4, no results of packet delivery ratio, throughput of the proposed system compared to those of Zigbee-based system were given. In addition, the comparisons of power consumption should be added.

Some minor issues are

The sentence "Because data ..." (lines 40~44) is difficult to understand.

In Fig. 1 and corresponding sentences, "meshed topology" should be changed to "mesh topology". Moreover, "Fonction" should be corrected.

In Lines 185, references [1] ~ [3] and [23] ~ [25] should be grouped.

In Lines 320, Fig. 1 was wrong indexed.

Author Response

Answers to Reviewer 3:

The reviewer’s comments are given in italic font: our point to point answers are given in bold font.

Open Review

English language and style

( ) Extensive editing of English language and style required
( ) Moderate English changes required
(x) English language and style are fine/minor spell check required
( ) I don't feel qualified to judge about the English language and style

Yes

Can be improved

Must be improved

Not applicable

Does the introduction provide sufficient background and include all   relevant references?

(x)

( )

( )

( )

Is the research design appropriate?

(x)

( )

( )

( )

Are the methods adequately described?

( )

(x)

( )

( )

Are the results clearly presented?

(x)

( )

( )

( )

Are the conclusions supported by the results?

( )

(x)

( )

( )

Comments and Suggestions for Authors

The reviewer appreciate the authors' effort put in this manuscript. However, there exist some following major concerns which need to me addressed:

The authors should emphasize more the contributions in the Introduction part of this manuscript by summarizing its contributions in a paragraph. It is not clear whether this manuscript is the first work to monitor outdoor fires based on an OCARI protocol or not.

The reviewer is right: the paper suffers from a concise form about this point. Especially, the originality of the presented work should be better pointed out than through the list [7-11] given in the introduction. We will add further comments (in red) in the revised version of the paper. For summarizing them here, no measurement system was designed with WSNs for measurement of fire properties, except our previous study [6]: WSNs are only used as binary (no fire/fire) detection system. The first attempt for providing a continuous multi-point measurement in fire with a WSN was officially with Zigbee [6]. That study revealed the limitation of irrelevant timestamping, that is overcome in the present work by the use of the OCARI stack.

In Section 3, detailed illustrations of the measurement devices and whole system should be given. Specifically, in Fig. 4, the author should show the dimension and explain the function of each component used in ADWAVE node. Moreover, its connection to themocouple and flex meters should be presented.

We thank the reviewer for this comment: indeed, we hesitated for choosing the level of details to give in the section “Experimental protocol”. Because of the large scope of this multidisciplinary paper –WSN software and hardware, heat measurements for fire, outdoor conditions…-, we finally chose to describe the conditioning of heat sensors with OCARI platform in the Annex B. This is not included in the main part of the text for a sake of readability.  Furthermore, we are not allowed to detail the whole architecture of the Adwave node given in fig.3 for evident commercial reasons.  We encourage the reviewer to contact directly Adwave for further information about the architecture. Finally, the details of the connections between analog transducers and nodes are explained in Annex B.

The authors should mention and discuss the heat resistance ability of

ADWAVE node and the ability of working stability in a fire.

This is a relevant question but it may address two different aspects:

A/ the ability to use an electronic device in a fire ensuring that its nominal operating conditions are respected thanks to thermal shields, leading to its stable behaviour

B/ the stability of the routing algorithm when the radio communications are disrupted by the fire.

Concerning A/, we have a 15 years knowledge on how protecting electronics systems embedded into a natural fire in the open (at real scale). This is less relative to the Adwave devices than to the thermal shield, composed of ceramic wood and Zetec-alu materials, we manufactured for each node. Furthermore, in [6], we observed the limitations of their resistance empirically. Thermal shields were observed to resist to an high intensity fire (10 kg/m² load of vegetal fuel, forming a 1m high vegetal litter and 4 m high flames resulting in level of heat radiation up to 35 kW/m²), but repetitions of fire tests destructed them. As a result, the embedded sensors were destroyed. The radio communications were stopped and the electronic components burnt. Before reaching to this irreversible situation, the rate of send-received messages slows down: the flow of validated messages through the WSN decreases as more as the fire size increases, certainly due to an increasing level of electromagnetic disruptions in a large scale fire.

This comment is included in the revised version of the paper (blue font).

If the question concerns the stability of the message routing protocol in the fire, we can just provide an a-priori answer because an exact one would require an in-situ control of the routing effectiveness. We observed in [6] that the rate of lost messages increases with the fire size but we have not investigated any diagnostic exhibiting the dependence of routing failures to the fire sizes.

Figs. 6 and 7 should be represented to increase their clarity. In the upper part, the setup experiment system cannot be seen. While in the lower parts, the authors should show the distance between sensor nodes, the wind direction.

The figures 6 and 7 are redrawn according to the recommendations of the reviewer.

It is stated in the Introduction that OCARI-based measurement system provides lower loss of radio messages compared with Zigbee-based measurement system. However, in Section 4, no results of packet delivery ratio, throughput of the proposed system compared to those of Zigbee-based system were given. In addition, the comparisons of power consumption should be added.

The reviewer is misleading: the Introduction didn’t stated a higher delivery ratio of messages with OCARI in comparison with Zigbee. We just mentioned (line 101) that the previous study published in Fire Technology [6] exhibited an increasing loss of radio messages with increasing fire size. Because of the convenient data timestamping allowed by the OCARI stack, the loss of messages becomes of minor importance in the ambition to replace a wired measurement system with a WSN. Seconds, the reviewer is right considering that the comparison of power consumption is of primary interest: but with the main aim of timestamping, the paper is already consequent (56 pages with Annexes). We cannot provide here the analysis of any other quantity in a sake of readability.  Furthermore, for an equivalent duration of fire experiments (about 300 s) presented in [6] as in that study, the enhanced performances in energy consumption can be illustrated by the size reduction of the load cells: a 6Ah Pb load cell per node is needed for the Zigbee Full Function Device Protocol (each node is a router) whereas a 1.2Ah load cell per node is sufficient for the OCARI FFD one.  The enhancement of the OCARI based solutions in term of energy consumption and compactness of each node (see fig.3 ) proceeds also from the size reduction of the load cell.

This is included in the revised version of the paper (orange font).

Some minor issues are:

The sentence "Because data ..." (lines 40~44) is difficult to understand.

Further explanations are given (green font in Introduction)

In Fig. 1 and corresponding sentences, "meshed topology" should be changed to "mesh topology". Moreover, "Fonction" should be corrected.

In Lines 185, references [1] ~ [3] and [23] ~ [25] should be grouped.

In Lines 320, Fig. 1 was wrong indexed.

These issues are corrected in the revised version of the paper.

Round  2

Reviewer 3 Report

The authors made sufficient revisions. The reviewer has no further technical comments. Hence, this manuscript is recommended to be published in Sensors journal.